# Irreversible Thermodynamics of Seawater Evaporation [†]

**Rainer Feistel** [1,2,*] [iD] **and Olaf Hellmuth** [2,3] [iD]

1  Leibniz Institute for Baltic Sea Research (IOW), 18119 Rostock, Germany
2  Leibniz-Sozietät der Wissenschaften zu Berlin e.V. (LS), 10117 Berlin, Germany; olaf@tropos.de
3  Leibniz Institute for Tropospheric Research (TROPOS), 04318 Leipzig, Germany
*  Correspondence: rainer.feistel@io-warnemuende.de
†  Dedicated to the 150th anniversary of the appearance of Gibbs' Fundamental Thermodynamic Relation.

**Abstract:** Under typical marine conditions of about 80% relative humidity, evaporation of water from the ocean is an irreversible process accompanied by entropy production. In this article, equations are derived for the latent heat of irreversible evaporation and the related nonequilibrium entropy balance at the sea surface. To achieve this, linear irreversible thermodynamics is considered in a conceptual ocean evaporation model. The equilibrium thermodynamic standard TEOS-10, the International Thermodynamic Equation of Seawater—2010, is applied to irreversible evaporation under the assumption of local thermodynamic equilibrium. The relevance of local equilibrium conditions for irreversible thermodynamics is briefly explained. New equations are derived for the mass flux of evaporation and for the associated nonequilibrium enthalpies and entropies. The estimated entropy production rate of ocean evaporation amounts to 0.004 W m$^{-2}$ K$^{-1}$ as compared with the average terrestrial global entropy production of about 1 W m$^{-2}$ K$^{-1}$.

**Keywords:** seawater; humid air; chemical potential; enthalpy; entropy; TEOS-10; local equilibrium; Onsager forces; entropy production

## 1. Introduction

*Iris in her rainbow garment lifted water, bringing fresh supplies to the clouds.*

*Ovid: Metamorphoses.*

The marine troposphere acts as a giant natural desalination plant. The evaporation of water from aqueous solutions is a fundamental process relevant to various disciplines such as meteorology, hydrology, climatology, physiology, or technology. However, its theoretical description, observation, measurement, and numerical modelling are still only insufficiently well understood and advanced. Edmond Halley (1687)[1] [1] was the first to estimate oceanic evaporation rates, finding that "the whole Mediterranean must lose in Vapour, in a Summers-day, at least 5280 Millions of Tons". John Dalton (1798)[2] [2] concluded from his experiments that the evaporation flux is driven by the liquid's vapor pressure by saying that "the quantity of any liquid evaporated in the open air is directly as the force of steam from such liquid".

Thermodynamically, evaporation from a liquid aqueous mixture such as seawater into a gaseous aqueous mixture such as humid air is a nonequilibrium process driven by the difference between the chemical potentials of water in the two phases. The resulting mass flux of pure water is irreversible and accompanied by an enthalpy flux ("latent heat") as well as a flux and additional production of entropy. In this paper, rigorous thermodynamic formulas are derived for the air–sea chemical potential difference, expressed in terms of relative fugacity, and for the evaporation enthalpy and entropy. These equations can be evaluated numerically using the open-access source-code library of the latest international geophysical standard, the Thermodynamic Equation of Seawater—2010 (TEOS-10, IOC et al. 2010 [3]). The related results may serve as the currently most accurate reference

values available for comparison with the various empirical and in part obsolete expressions implemented in some climate models. Prior to TEOS-10, correspondingly mutually consistent empirical equations for the requisite fugacities, enthalpies, and entropies had not been available from any geophysical standards, such as the 1980 International Equation of State of Seawater (Unesco 1983, Feistel 2018, Smythe-Wright et al., 2019) [4–6].

Globally, the dominating energy source for the dynamics and the warming of the atmosphere is *not* the terrestrial thermal radiation trapped by greenhouse gases; rather, it is the latent heat of water vapor evaporated from the oceans (Feistel and Hellmuth 2021) [7]. "The by far largest part of heat conveyed to the air is in the form of latent heat during subsequent condensation along with cloud formation. . . . The heat budget over the sea is mainly controlled by the latent heat released to the air"[3]. "For the atmosphere, the globally averaged net radiative cooling approximately balances the globally averaged latent-heat release. The latent heat is supplied by the evaporation of water from the surface"[4] [9]. "This way, the heat released to the air in latent form is larger by a multiple than the [sensible] heat transferred immediately to the air"[5]. The devastating power of evaporation from the exceptionally warm Mediterranean has recently become evident again by an unprecedented flooding of Slovenia and atypical gale force winds in summer over the Baltic Sea, caused by the cyclone Petar (German name: Zacharias) in August 2023, and also mid-September by torrential rain poured over Greece and Libya by "Daniel", the deadliest Mediterranean tropical-like cyclone in recorded history.

Climate models estimate the ocean–atmosphere heat flux to within a relatively poor uncertainty of at least $5 - 10 \, \mathrm{Wm^{-2}}$ (Josey et al., 2013; Rhein et al., 2013; Cronin et al., 2019) [10–12], a range that is 1000 times as large as the mean atmospheric warming rate of just $0.005 \, \mathrm{Wm^{-2}}$ (Feistel 2015; Gorfer 2022; von Schuckmann et al., 2023) [13–15]. This suggests the conclusion that, strictly speaking, current climate models are unable to estimate with requisite significance the rate of global warming of the troposphere. Within the given uncertainty of the model prediction, the atmosphere either may warm up much faster than observed or may even cool down. Moreover, although "the climate of the Earth is ultimately determined by the temperatures of the oceans"[6], "most CMIP6 [Coupled Model Intercomparison Project Phase 6] models fail to provide as much heat into the ocean as observed"[7]. Among other reasons for this mismatch, a biased parameterization of evaporation fluxes may also contribute (Feistel and Hellmuth 2023) [18]. The observationally and theoretically complex and challenging process of oceanic evaporation warrants enhanced efforts for improving its physical description.

"It is one of the priorities of the World Climate Research Program (WCRP) to improve the accuracy of surface fluxes for climate studies to within »a few W/m²« and 10 W/m² for individual flux components and the large scale net heat fluxes, respectively. . . . These requirements impose challenges including the development of new parameterizations, achievement of global and regional heat budget closure, reducing sampling uncertainties, and better scaling parameters for surface flux estimates" (Bentamy et al., 2017 [19]: p. 196). The new equations derived in this paper are intended to contribute to this aim.

In Figure 1, a typical measured microstructure of a temperature profile immediately above the sea surface is shown. A laminar layer in the submillimeter range is distinguished by its vertical gradient, while the air column above appears to be isothermal with turbulent fluctuations. The thickness of the laminar layer is decreasing with the wind speed (Hupfer et al., 1975) [20]. Near-surface humidity profiles up to 40 m height were reviewed by Avery (1972) [21]. Recent near-surface data are often obtained from remote sensing and do not resolve the vertical microstructure (Gao et al., 2019) [22].

In this paper, the irreversible thermodynamics of seawater evaporation is described theoretically in a simplified conceptual model as depicted in Figure 2. Similar models had already been employed previously by Schmidt (1915), Wüst (1920), Sverdrup (1936), Montgomery (1940), Albrecht (1940), Sellers (1960), Budyko (1963), or Debski (1966) [8,24–30]. For the evaporation mass flux $J_W$ across the laminar layer, Sverdrup (1936 [26]: Equation 10 therein) used the equation

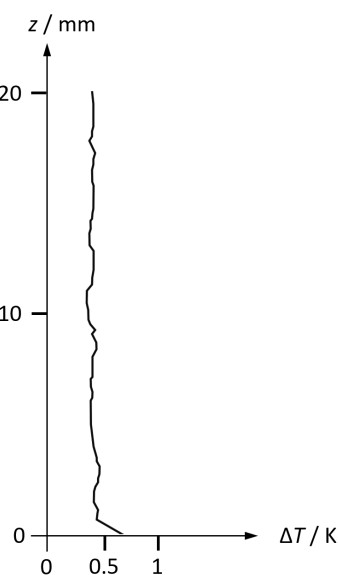

**Figure 1.** Microstructure of temperature in the atmospheric layer immediately above the sea surface, measured on 15 April 1975 at the Caspian Sea (Foken et al., 1978) [23]. To the authors' knowledge, similar measured profiles of relative humidity at the open sea surface do not exist in the literature. This figure provides insight into the structure of the immediate boundary layer of some millimeter thickness. In the millimeter range, there is a thermal skin effect visible with a linear gradient, while the profile above is homogeneous due to turbulent mixing with related fluctuations. Assuming that the vertical profile of specific humidity is homogeneous (as a result of water vapor conservation during turbulent mixing), the constant vertical temperature profile indicates also a constant profile of relative humidity (data courtesy of Thomas Foken, priv. comm).

$$J_{\text{W}} = -\delta \frac{e - e^{\text{sat}}}{\Delta z}. \tag{1}$$

Here, $\delta$ is an empirical coefficient expressing the characteristic diffusion time, $\Delta z$ is the layer thickness, $e^{\text{sat}}$ is the saturation vapor pressure of water/seawater, and $e$ is the partial pressure of water vapor of humid air above the sea surface. Montgomery (1940 [27]: Equation 24 therein) considered specific humidity $q$ in place of vapor pressure $e$ in Equation (1), an approximation still being used by various modern climate models (Gill 1982; Stewart 2008; Rapp 2014) [16,31,32]. Both forms are known as *Dalton equation* today.

Evaporation occurs successively in two distinct physical steps. First, thermally fast water molecules escape from the condensed phase (water, seawater, ice, moist soil), and second, they are carried away in the gas phase by molecular or turbulent diffusion. The transfer of water across the phase boundary is driven by the difference between the chemical potentials of water in the two phases. The systematic loss of the fastest molecules of the condensed phase is lowering the skin temperature (Saunders 1967; Schluessel et al., 1990; Zülicke and Hagen 1998; Katsaros 2001; Zülicke 2005) [33–37], and the implied loss of heat (i.e., enthalpy) takes the form of latent heat added to the humid air.

Chemical potentials were introduced by J. Willard Gibbs (1878) [38] as chemical equilibrium conditions of composite systems. They cannot be measured directly, and for seawater and humid air, they have become available numerically for the first time through TEOS-10, the *Thermodynamic Equation of Seawater—2010* (IOC et al., 2010; Feistel et al., 2010, 2016; Feistel 2012, 2018) [3,5,39–41]. In irreversible thermodynamics (Landau and Lifschitz 1966; Glansdorff and Prigogine 1971; Subarew 1976; DeGroot and Mazur 1984) [42–45], the driving force for a mass flux is the gradient $X = \nabla(\mu/T)$ of the chemical potential $\mu$ divided by the temperature $T$. This applies to seawater evaporation as well (Kraus and Businger 1994) [46]. If the evaporation flux is rigorously expressed this way, the historical Dalton Equation (1) can be derived as a certain approximation thereof (Feistel and Hellmuth 2023) [18].

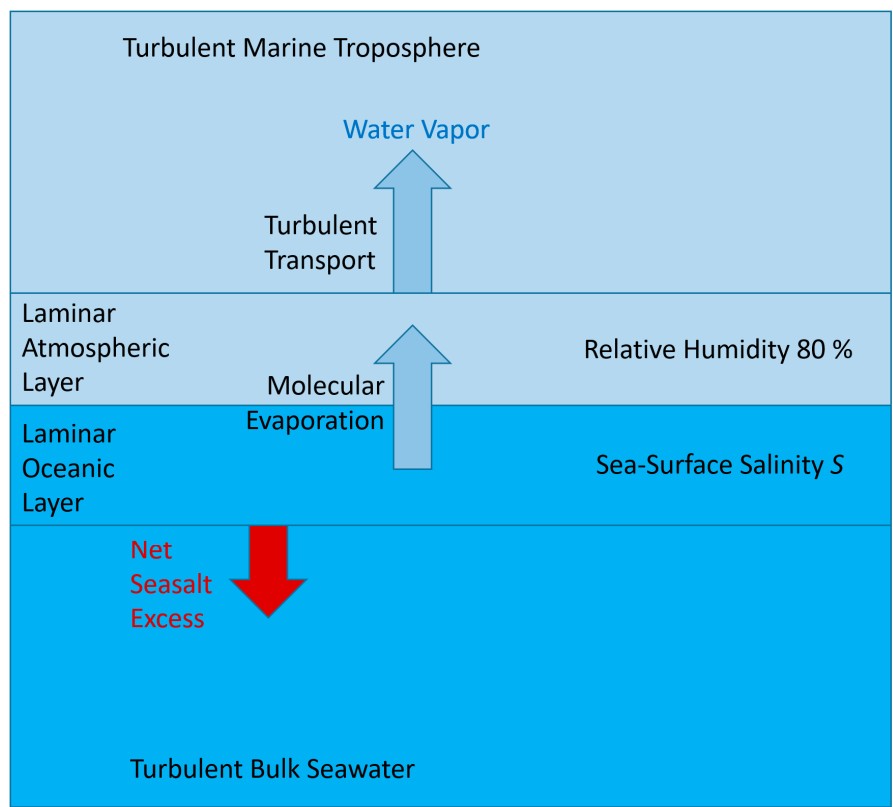

**Figure 2.** Schematic of a conceptual model of seawater evaporation. The ocean–atmosphere interface is embedded in laminar layers on both the liquid and the gas sides. These layers are assumed to be sufficiently thin so that molecular transport may establish homogeneous distributions of specific humidity, salinity, and temperature. Outside those layers, humid air and seawater may be turbulently mixed.

TEOS-10 was recommended as an international standard by IOC-UNESCO (2009) [47] in Paris and by IUGG (2011) [48] in Melbourne. It is defined and explained in the TEOS-10 Manual (IOC et al., 2010) [3], which, together with several background papers, is freely available from the internet at http://www.TEOS-10.org. Empirical equations with their numerical coefficients are defined in a series of IAPWS documents (IAPWS AN6-19 2016) [49]. Functions for numerous thermodynamic properties are implemented as an open-source code in the SIA ("Sea–Ice–Air") and the GSW ("Gibbs Seawater") libraries (Wright et al., 2010) [50], also available from in TEOS-10 web page. In an appendix of Feistel et al. (2022) [51], an additional explicit TEOS-10 code is provided for the calculation of relative fugacity, a real gas alternative to conventional relative humidity, as defined mathematically by Feistel and Lovell-Smith (2017) [52]. Simplified TEOS-10 equations for application at the sea surface are reported in an appendix of Feistel and Hellmuth (2023) [18].

This paper is structured as follows: Section 2 provides a short introduction to nonequilibrium thermodynamics and the role that *local equilibrium* plays in it. Section 3 derives the thermodynamic equation for the evaporation flux, expressed in terms of relative fugacity of water vapor in humid air, showing that its empirical rate constant systematically deviates from Dalton equations used in numerical climate models. Section 4 proves that the formula for the latent heat of reversible evaporation remains valid also for the irreversible nonequilibrium situation observed naturally at the oceans. By contrast, the entropy of irreversible evaporation, Section 5, includes an additional entropy production term proportional to the distance from the equilibrium of the ocean–atmosphere interface, which may as well be expressed in terms of relative fugacity. A summary concludes this paper, and the Nomenclature provides a list of the formula symbols used.

In a steady-state regime, the water vapor entering the laminar layer by evaporation (see Figure 2) is conveyed away at the same rate in the layers above (Sverdrup 1936) [26]. While this paper is focused on the immediate air–sea interface, a wealth of related literature is available regarding the parameterization of the turbulent vertical water vapor flux in the near-surface layer of the troposphere in terms of wind speed and water surface roughness. A brief introduction and review, in particular concerning the Monin–Obukhov similarity theory (MOST) and a compilation of vapor-pressure- and specific-humidity-based Dalton coefficients from the literature, is separately added to this paper in a supplement.

## 2. Irreversible Thermodynamics and Local Equilibrium

Numerous textbooks on irreversible thermodynamics are available (Landau and Lifschitz 1966; Glansdorff and Prigogine 1971; Falkenhagen et al., 1971; De Groot and Mazur 1984; Feistel and Ebeling 2011) [42,43,45,53,54]; however, most of these do not cover irreversible processes in multiphase composite systems or at phase boundaries, such as those required for clouds or for the ocean–atmosphere interface. Here is a brief introduction.

The heat content of a given sample with a *temperature T* is described thermodynamically by two different quantities, *entropy N* and *enthalpy H*. Note that, here, $N$ is used for entropy rather than $S$ to avoid confusion with the traditional ocean salinity variable. The relations of $N$ and $H$ to heat are evident from their relations to the *isobaric heat capacity*, $C_p$, either by

$$C_p = T \left( \frac{\partial N}{\partial T} \right)_p \tag{2}$$

or by

$$C_p = \left( \frac{\partial H}{\partial T} \right)_p, \tag{3}$$

both being defined at constant pressure, $p$. The difference between the two measures of heat is the *Gibbs energy* (also known as *free enthalpy* or *Gibbs free energy*):

$$G = H - TN. \tag{4}$$

These extensive quantities may vary over time $t$ due to internal processes (subscript i) or exchange across the boundary (subscript e):

$$\frac{dN}{dt} = \frac{d_i N}{dt} + \frac{d_e N}{dt}, \tag{5}$$

$$\frac{dH}{dt} = \frac{d_i H}{dt} + \frac{d_e H}{dt}, \tag{6}$$

$$\frac{dG}{dt} = \frac{d_i G}{dt} + \frac{d_e G}{dt}. \tag{7}$$

According to the *second law of thermodynamics*, reversible processes are defined by zero *entropy production*, $P \equiv \frac{d_i N}{dt} = 0$, and, by contrast, irreversible processes by $\frac{d_i N}{dt} > 0$, while processes with $\frac{d_i N}{dt} < 0$ are physically impossible. After an irreversible change, the system may never return to its former state without exchange with its surroundings because the entropy, once produced, cannot be destroyed again but may only be exported to the outside world. Note that entropy production $P > 0$ does not necessarily imply a warming but may as well be associated with structural relaxation (Kirkaldy 1965 [55]; Landau and Lifschitz 1966 [56]: Equation (13,7) therein; Feistel 2019 [57]; Zivieri 2023 [58]), such as weathering or mixing.

Entropy production can always be expressed as a bilinear sum:

$$P = \sum_k J_k X_k. \tag{8}$$

Here, $X_k$ are the *Onsager forces* describing the deviation from the equilibrium, such as temperature or concentration gradients. $J_k$ are the associated fluxes driven by those forces. At the equilibrium, both $X_k$ and $J_k$ vanish. Reversible processes may occur at $X_k \neq 0$ but $J_k = 0$. As long as the forces are sufficiently weak, fluxes may be represented as linear functions of the forces,

$$J_i = \sum_k \Omega_{ik} X_k, \tag{9}$$

expressing the rationale of *linear irreversible thermodynamics*. If the *Onsager coefficients* $\Omega_{ik}$ do not depend on the forces, the *Prigogine theorem* holds, meaning that steady states establish minimum entropy production (Prigogine and Wiaume 1946) [59]. This theorem quantifies the principle of Le Chatelier and Braun that fluxes always tend to reduce the forces. Under these conditions, instabilities, spontaneous self-organization, or chaos is excluded.

Expressing the *first law of thermodynamics*, reversible and irreversible processes at constant $T$ and $p$, such as evaporation or condensation, are characterized by the conservation of enthalpy, $\frac{d_i H}{dt} = 0$. Under these conditions, the second law implies $\frac{d_i G}{dt} = 0$ for reversible and $\frac{d_i G}{dt} < 0$ for irreversible processes. According to Gibbs (1878) [38], $G$ may be expressed as the sum

$$G = \sum_k \mu_k m_k \tag{10}$$

of the mass-specific chemical potentials $\mu_k$ and the masses $m_k$ of the sample's constituents. For a pure substance with the mass $m$, its chemical potential equals its *specific Gibbs energy*, $g \equiv G/m = \mu$.

In meteorology and oceanography, additional practical measures of the heat content are *potential temperature* $\vartheta$ and *conservative temperature* $\Theta$, as defined, respectively, in terms of entropy, $N(T, p)$, and enthalpy, $H(T, p)$. Potential temperature $\vartheta(T, p)$ of a parcel with in situ temperature $T$ at pressure $p$ is implicitly given by the actual temperature of that parcel after lifting or lowering it isentropically (at $N = const.$) in the gravity field to the surface pressure $p_0$:

$$N(T, p) = N(\vartheta, p_0). \tag{11}$$

This definition works for the atmosphere and ocean as well. This way, $\vartheta$ is some equivalent measure of entropy, mainly designed for reversible excursion processes. Equilibrium entropy may be determined experimentally up to an arbitrary constant, which may only be concluded from statistical models (Planck 1906; Feistel and Wagner 2005, 2006; Feistel 2019) [57,60–62] or by formal specification of an arbitrary reference state (Wagner and Pruß 2002; Feistel et al., 2008; IOC et al., 2010; Feistel 2018) [3,5,63,64]. While the definition (11) is independent of the choice of this arbitrary constant, other common definitions are ambiguous in this respect, such as that of Gill (1982 [31]: Equation 3.7.7 therein). Potential temperature is less suited as a conservative tracer if the parcel undergoes irreversible internal mixing by which additional entropy is produced. In such a case, the potential temperature is formally rising even though no extra heat has entered the parcel. To reduce this and other non-conservative effects, *potential enthalpy H\** is defined (McDougall 2003; IOC et al., 2010; Graham and McDougall 2013; McDougall et al., 2023) [3,65–67] by

$$H^*(T, p) = H(\vartheta(T, p), p_0). \tag{12}$$

Conservative temperature is the specific potential enthalpy conveniently scaled to a temperature unit, $c\,\Theta = H^*/m$. In oceanography, this conversion constant is specified as the specific isobaric heat capacity of seawater at the ocean standard state. Being defined so far for oceanography only, the use of this measure in the atmosphere would be straightforward.

Geophysical fluids, such as the troposphere or ocean, are typically in nonequilibrium states but may successfully be described thermodynamically under the assumption of *local equilibrium* as introduced by Ilya Prigogine (1947, 1978) [68,69]. This means that sufficiently small volumes (termed "cells" here) exist in which the particle velocities obey a statistical Maxwell distribution so that a local temperature may reasonably be assigned

to such a cell. A well-defined temperature is an indispensable precondition for proper definitions of entropy, Gibbs energy, and chemical potentials. As a consequence of local equilibrium, entropy production does not occur within a cell but only due to fluxes between adjacent cells whose spatial property differences then constitute the Onsager forces. Usual equilibrium thermodynamics applies to local equilibrium cells, described by their specific entropy, $\eta = N/m$; specific enthalpy, $h = H/m$; specific volume, $v = V/m$; mass fractions, $w_k = m_k/m$; and specific internal energy, $u = U/m$.

In statistical equilibrium thermodynamics, the so-called microcanonical ensemble provides a fundamental thermodynamic potential in the form of entropy, $N(U, V, \boldsymbol{m})$, as a function of internal energy, $U$; volume, $V$; and masses of chemical species, $\boldsymbol{m}$ (see Equation (19) below). All arguments of this potential are extensive quantities subject to conservation laws. Specific entropy expressed as a function of specific energy, $u$; specific volume, $v$; and mass fractions of the constituents, $w_k, k = 1, 2, \ldots$,

$$\eta = \eta(u, v, w_k), \tag{13}$$

obeys Gibbs' fundamental equilibrium relation in the form (Fofonoff 1962 [70]; Glansdorff and Prigogine 1971 [43]: Equation 2.15 therein)

$$\frac{\mathrm{d}\eta}{\mathrm{d}t} = \frac{1}{T}\frac{\mathrm{d}u}{\mathrm{d}t} + \frac{p}{T}\frac{\mathrm{d}v}{\mathrm{d}t} - \sum_k \frac{\mu_k}{T}\frac{\mathrm{d}w_k}{\mathrm{d}t}. \tag{14}$$

Here, because of the local equilibrium, the cell's entropy change $\mathrm{d}\eta/\mathrm{d}t = \mathrm{d_e}\eta/\mathrm{d}t + \mathrm{d_i}\eta/\mathrm{d}t$ is caused exclusively by exchange processes through the cell's border, $\mathrm{d_e}\eta/\mathrm{d}t$, and does *not* include any internal entropy production contributions, $\mathrm{d_i}\eta/\mathrm{d}t = 0$. Hence, the sign of the derivative $\mathrm{d}\eta/\mathrm{d}t$ in Equation (14) is not restricted in any way by the second law.

If, for example, two cells of the same mass, each at local equilibrium but away from mutual equilibrium, are included in an isolated parcel (without any exchange with its surrounding), $\mathrm{d_e}\eta/\mathrm{d}t = 0$ (see Figure 3), the change of its total entropy is confined to the entropy production rate inside the parcel, $\mathrm{d}\eta/\mathrm{d}t = \mathrm{d_i}\eta/\mathrm{d}t \geq 0$. Adding up the entropies, Equation (14), of the two cells, while accounting for the conservation laws of energy, $\mathrm{d}u_1 = -\mathrm{d}u_2$; of volume, $\mathrm{d}v_1 = -\mathrm{d}v_2$; and of matter of each constituent, $\mathrm{d}w_1 = -\mathrm{d}w_2$, the parcel's total entropy will increase at the rate

$$\frac{\mathrm{d}\eta}{\mathrm{d}t} = \frac{\mathrm{d_i}\eta}{\mathrm{d}t} = \frac{\mathrm{d}\eta_1}{\mathrm{d}t} + \frac{\mathrm{d}\eta_2}{\mathrm{d}t} = \left(\frac{1}{T_1} - \frac{1}{T_2}\right)\frac{\mathrm{d}u_1}{\mathrm{d}t} + \left(\frac{p_1}{T_1} - \frac{p_2}{T_2}\right)\frac{\mathrm{d}v_1}{\mathrm{d}t} - \left(\frac{\mu_1}{T_1} - \frac{\mu_2}{T_2}\right)\frac{\mathrm{d}w_1}{\mathrm{d}t} \geq 0. \tag{15}$$

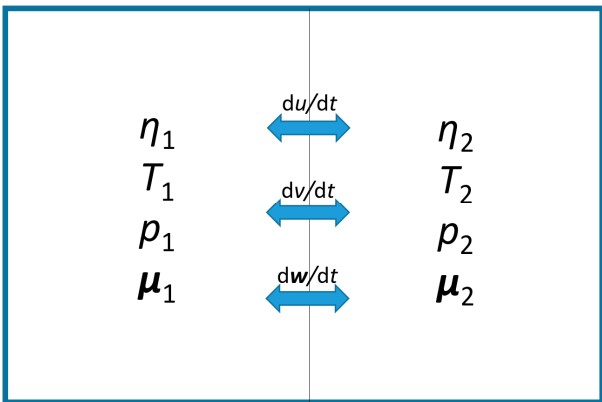

**Figure 3.** Schematic of an isolated nonequilibrium parcel consisting of two different local equilibrium cells with equal masses, subject to mutual exchange. Accordingly, for the entire parcel, the second law takes the form $\frac{\mathrm{d}\eta}{\mathrm{d}t} = \frac{\mathrm{d_i}\eta}{\mathrm{d}t} = \frac{\mathrm{d}\eta_1}{\mathrm{d}t} + \frac{\mathrm{d}\eta_2}{\mathrm{d}t} \geq 0$, and the first law is $\frac{\mathrm{d}u}{\mathrm{d}t} = 0 = T\frac{\mathrm{d_e}\eta}{\mathrm{d}t} - p\frac{\mathrm{d}v}{\mathrm{d}t} + \sum_k \mu_k \frac{\mathrm{d}w_k}{\mathrm{d}t} \leq T\frac{\mathrm{d}\eta}{\mathrm{d}t} - p\frac{\mathrm{d}v}{\mathrm{d}t} + \sum_k \mu_k \frac{\mathrm{d}w_k}{\mathrm{d}t}$.

Here, differences in intensive quantities between adjacent cells are the thermodynamic forces, $\Delta\frac{1}{T}$, $\Delta\frac{p}{T}$, and $-\Delta\frac{\mu}{T}$, which drive, in the direction of relaxation to mutual equilibrium, the particular exchange fluxes of extensive quantities per unit mass; internal energy, $\frac{du}{dt}$; occupied volume, $\frac{dv}{dt}$; and matter, $\frac{dw}{dt}$, across the boundary separating the cells. Similarly, in the general case of an inhomogeneous nonequilibrium parcel, corresponding spatial gradients between its neighboring cells with respect to the related partial differentials of entropy (14) constitute the appropriate three-dimensional Onsager force vectors (Glansdorff and Prigogine 1971) [43],

$$\boldsymbol{X}_u \equiv \nabla \left( \frac{\partial \eta}{\partial u} \right)_{v,w_k} = \nabla \frac{1}{T} \, , \tag{16}$$

$$\boldsymbol{X}_v \equiv \nabla \left( \frac{\partial \eta}{\partial v} \right)_{u,w_k} = \nabla \frac{p}{T} \, , \tag{17}$$

$$\boldsymbol{X}_k \equiv \nabla \left( \frac{\partial \eta}{\partial w_k} \right)_{u,v,w_{i \neq k}} = -\nabla \frac{\mu_k}{T} \, . \tag{18}$$

Fluxes driven by these forces give rise to irreversible entropy production among the spatial arrangement of cells. Note that because of the identity $\sum_k w_k \equiv 1$, one of Equation (18) is not independent of the remaining ones and may be omitted.

At this point, for clarity and honesty, it may be important to mention that for the description of entropy production in terms of irreversible forces and fluxes, the existence of local equilibria and their temperatures, as argued above, is sufficient but not necessary. The fundamental statistical definition of entropy,

$$N = k \log W, \tag{19}$$

was first proposed by Max Planck (1906) [60] as a mathematical tool supporting his discovery of energy quanta of thermal radiation. Here, $W$ is the number of microstates that are consistent with the given macroscopic boundary conditions. At that time, Planck attributed the origin of this formula to Boltzmann (1877, 1896) [71,72], so that Equation (19) became generally known as the "Boltzmann entropy" with the "Boltzmann constant" $k$, which has recently become the key to the current SI definition of kelvin (BIMP 2019) [73]. The famous entropy formula is engraved at Boltzmann's monument in Vienna (Figure 4).

However, the work of Boltzmann (1877, 1896) [71,72] does not include Equation (19) in an explicit form. In a text written in 1945 that appeared only shortly after his death, Planck (1948) [74] revealed his own authorship. "Als Resultat . . . stellte sich heraus, . . . daß dabei *k* die sog. absolute Gaskonstante vorstellt . . . Sie wird öfters verständlicherweise als die *Boltzmannsche* Konstante bezeichnet. Dazu ist allerdings zu bemerken, daß *Boltzmann* diese Konstante weder jemals eingeführt noch meines Wissens überhaupt daran gedacht hat, nach ihrem numerischen Wert zu fragen. . . . Was nun die Größe *W* anbetrifft, so erwies es sich, um diese Größe als Wahrscheinlichkeit deuten zu können, als notwendig, eine neue universelle Konstante einzuführen, die ich mit *h* bezeichnete und . . . das elementare Wirkungsquantum nannte. Damit war also das Wesen der Entropie im Sinne *Boltzmanns* auch in der Strahlung festgestellt"[8].

Let the state of an arbitrary physical object, which consists of many elements, sufficiently be well characterized by some set $\boldsymbol{\zeta}$ of macroscopic properties. If a number $W(\boldsymbol{\zeta})$ of different random microstates of the elements exists to each particular macrostate $\boldsymbol{\zeta}$ of the object, then an entropy value $N(\boldsymbol{\zeta})$ may be associated with that macrostate according to Planck's entropy formula (19). This entropy is statistically well defined regardless of the existence of local equilibria or temperatures. Note, however, that by $T^{-1} = dN/dU$, a formal temperature can be assigned to such an object or its parts as far as the object's energy $U$ is a well-defined physical quantity (Planck 1906 [60], Landau and Lifschitz 1966 [56]: §9 therein). On the other hand, $W$ is the number of microstates of a given system, which is independent of the actual way those microstates are occupied by the system; in other words, $W$, as well as $N$, takes the same value regardless of describing an equilibrium or

a nonequilibrium state. Therefore, the value of Planck's entropy $N$, Equation (19), of a closed system does not increase during internal irreversible processes (Feistel 2019) [57], in contrast with Equation (15).

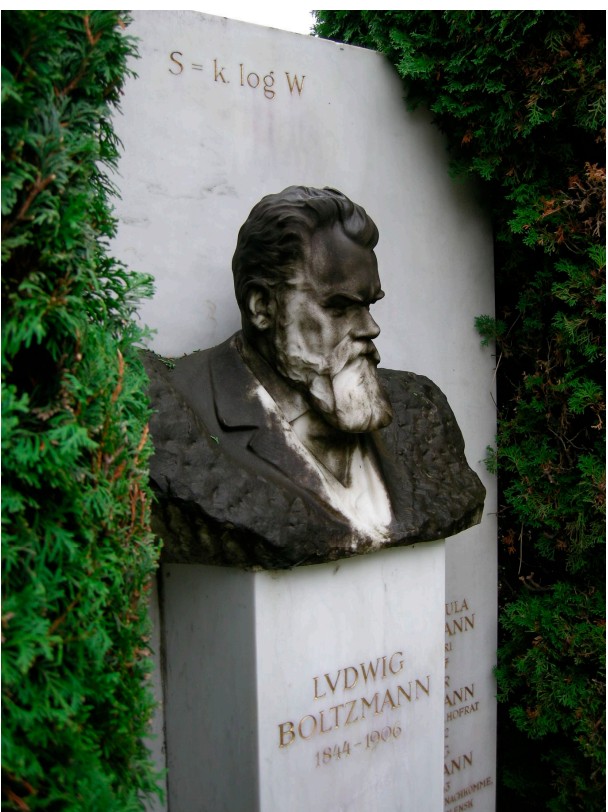

**Figure 4.** Equation of statistical entropy, ascribed to Ludwig Boltzmann, as engraved at the Vienna Central Cemetery. This fundamental entropy definition does not presuppose the existence of local equilibria or temperatures. Photo taken in October 2010.

If the object is isolated from its environment, the latter cannot affect its entropy, $d_e N/dt = 0$. If the object's state is changing anyway, this change is subject to the second law in the form of the bilinear sum,

$$d_i N/dt = P = \boldsymbol{J}\,\boldsymbol{X} \geq 0, \tag{20}$$

where the flux vector is $\boldsymbol{J} = d\boldsymbol{\xi}/dt$ and the conjugate forces are $\boldsymbol{X} = \partial N/\partial \boldsymbol{\xi}$ (Landau and Lifschitz 1966; Feistel and Ebeling 2011) [42,54]. So far, to the authors' knowledge, no system could be demonstrated to violate the inequality (20). Equations (15)–(18) represent just a special case of this fundamental concept, as applied here to oceanic evaporation. The statistical entropy (19) is relevant as residual Pauling entropy at the zero point, such as of ice Ih, and for other frozen structures, such as information carriers (Feistel 2019) [57].

Returning to geophysics, in the gravity field, fluids become vertically stratified. At equilibrium, the fluid has constant vertical profiles of the temperature and of each chemical potential, while entropy and concentrations possess individual vertical gradients (Guggenheim 1949; Landau and Lifschitz 1966) [56,75]. At a perfectly (turbulently) mixing state, by contrast, entropy and concentrations have vertically constant profiles, while temperature has an adiabatic gradient (McDougall and Feistel 2003; Feistel and Ebeling 2011) [54,76] together with the chemical potentials (Feistel and Hagen 1994; Feistel and Feistel 2006) [77,78]. The observed uniform chemical compositions of environmental dry air and sea salt do in fact result from that mixing. The mixing state is a nonequilibrium state that permanently produces entropy (Feistel 2011; Feistel and Ebeling 2011) [54,79] because of nonvanishing

fluxes (16)–(18) between fluid cells in close contact after they arrived adiabatically from different pressure levels, and due to hydrodynamic viscous friction.

## 3. Onsager Force and Flux of Evaporation

Previously, oceanographers such as Wüst (1920) [25], Sverdrup (1936) [26], and Albrecht (1940) [8] used the "vapor pressure jump" at the sea surface as the driving force for evaporation, in agreement with Dalton (1798) [2]. In modern irreversible thermodynamics, this jump is more correctly expressed by the difference between the chemical potentials of water in seawater and in humid air (Kraus and Businger 1994) [46], such as by the Onsager force (18) across the phase boundary (Doney 1994) [80].

Having been numerically unavailable in a sufficiently consistent way before 2010, more recently, those chemical potentials may be derived from the Gibbs potential of seawater, the specific Gibbs energy $g^{SW}(S, T, p) = G^{SW}/m$, and from that of humid air, $g^{AV}(A, T, p) = G^{AV}/m$, provided by TEOS-10, the Thermodynamic Equation of Seawater—2010 (IOC et al., 2010; Feistel 2018; Feistel and Hellmuth 2023) [3,5,18], the current international geophysical standard for thermodynamic properties of seawater, ice, and humid air. The Gibbs potentials do not depend on the respective sample masses, $m$. Their composition variables are absolute salinity, $S$; the mass fraction of dissolved salt in seawater; and the mass fraction $A$ of dry air in humid air. The chemical potential $\mu_W^{SW}$ of water in seawater with a solute mass $m_S = Sm$ and a water mass $m_W = m - m_S$ is calculated from

$$\mu_W^{SW} = \left(\frac{\partial G^{SW}}{\partial m_W}\right)_{m_S, T, p} = g^{SW} - S\left(\frac{\partial g^{SW}}{\partial S}\right)_{T, p} \tag{21}$$

and the chemical potential $\mu_V^{AV}$ of water vapor in humid air with a dry air mass $m_A = Am$ and a water vapor mass $m_V = m - m_A$ from

$$\mu_V^{AV} = \left(\frac{\partial G^{AV}}{\partial m_V}\right)_{m_A, T, p} = g^{AV} - A\left(\frac{\partial g^{AV}}{\partial A}\right)_{T, p}. \tag{22}$$

The chemical potential of water in seawater may formally be expressed by the fugacity, $f_W^{SW}$, in the form (Guggenheim 1949; Prausnitz et al., 1999) [75,81]

$$\mu_W^{SW} = \mu_W^{id}(T, x_W p) + R_W T \ln \frac{f_W^{SW}(x_W, T, p)}{x_W p}. \tag{23}$$

Here, the ideal gas chemical potential is defined asymptotically by the low pressure limit of $\mu_W$, using an arbitrary auxiliary reference pressure $p_0$,

$$\mu_W^{id}(T, p) = R_W T \ln \frac{p}{p_0} + \lim_{p \to 0}\left\{\mu_W(T, p) - R_W T \ln \frac{p}{p_0}\right\}. \tag{24}$$

The specific gas constant, $R_W = R/M_W$, is the molar gas constant, $R$, divided by the molar mass of water, $M_W$. Denoting by $M_S$ the molar mass of sea salt (Millero et al., 2008) [82], the mole fraction $x_W$ of water in seawater is

$$x_W(S) = \left[1 + \frac{SM_W}{(1-S)M_S}\right]^{-1}. \tag{25}$$

Similar to Equation (23), the chemical potential of water vapor in humid air is

$$\mu_V^{AV} = \mu_V^{id}(T, x_V p) + R_W T \ln \frac{f_V^{AV}(x_V, T, p)}{x_V p} \tag{26}$$

with the mole fraction $x_V$,

$$x_V(A) = \left[1 + \frac{AM_W}{(1-A)M_A}\right]^{-1}, \tag{27}$$

of water vapor. The molar mass of dry air is denoted by $M_A$. The fugacity of water vapor in humid air is $f_V^{AV}$ (Feistel et al., 2015) [83].

By its definition (26), the fugacity of an ideal gas equals its partial pressure, $f_V^{id} = x_V p$. The ideal gas chemical potentials $\mu_V^{id}$ of water vapor in humid air and $\mu_W^{id}$ of liquid water in seawater are identical functions describing the water substance so that their difference is, according to Equation (24),

$$\mu_V^{id}(T, x_V p) - \mu_W^{id}(T, x_W p) = R_W T \ln \frac{x_V}{x_W}. \tag{28}$$

The resulting difference

$$\mu_V^{AV}(A, T) - \mu_W^{SW}(S, T) = R_W T \ln \frac{f_V^{AV}(x_V, T, p)}{f_W^{SW}(x_W, T, p)} \tag{29}$$

is regarded as the *affinity of vaporization* (Kraus and Businger 1994 [46]: p. 42 therein).

Assuming the same temperature on both sides of the air–sea interface with thickness $\Delta z$, the evaporation force (18) follows from (29) to be

$$X_W = -\frac{R_W}{\Delta z} \ln \frac{f_V^{AV}(x_V, T, p)}{f_W^{SW}(x_W, T, p)}. \tag{30}$$

For mole fractions close to unity, $x_W \approx 1$, the Lewis fugacity rule (Prausnitz et al., 1999 [81]: Section 5.1 therein) holds that

$$f_W^{SW}(x_W, T, p) \approx x_W f_W^{SW}(1, T, p) = \left[1 + \frac{SM_W}{(1-S)M_S}\right]^{-1} f_V^{AV}(x_V^{sat}, T, p). \tag{31}$$

Here, the fugacity of water vapor in saturated humid air, $f_V^{AV}(x_V^{sat}, T, p)$, equals that of pure liquid water, $f_W^{SW}(1, T, p)$. The resulting equation for the evaporation mass flux from the sea surface with salinity $S$ is

$$J_W = D_f \left\{ \ln\left[1 + \frac{SM_W}{(1-S)M_S}\right] - \ln\psi_f(x_V, T, p) \right\}. \tag{32}$$

The relative fugacity, $\psi_f$, of water vapor in humid air is defined by (Feistel and Lovell-Smith 2017) [52]

$$\psi_f(x_V, T, p) = \frac{f_V^{AV}(x_V, T, p)}{f_V^{AV}(x_V^{sat}, T, p)}. \tag{33}$$

The Onsager coefficient of evaporation is $\Omega_{WW}$, and the fugacity-based empirical mass transfer coefficient is $D_f = \Omega_{WW} R_W / \Delta z$.

In ideal gas approximation of the fugacities, $f_V^{id} = x_V p = e$, that is,

$$\psi_f \approx \frac{x_V}{x_V^{sat}} = \frac{e}{e^{sat}} \approx \frac{q}{q^{sat}}, \tag{34}$$

the evaporation flux (32) is driven by the ratio of vapor pressures $e$ or of specific humidities, $q = 1 - A$, in contrast with their difference,

$$\widetilde{J_W} = D_q \left\{ \left(1 - \frac{M_W}{M_S}S\right) q^{sat}(T) - q \right\} = D_q q^{sat}(T) \left\{ 1 - \frac{M_W}{M_S}S - \frac{q}{q^{sat}(T)} \right\}, \tag{35}$$

in the Dalton equation of the form of Equation (1) or (35) typically used in numerical climate models with a constant empirical value of $D_q$ (Debski 1966; Baumgartner and Reichel 1975; Stewart 2008) [30,32,84].

Note that $1 - \frac{M_W}{M_S} S \approx 0.98$ for ocean salinities $S \approx 35 \text{ g kg}^{-1}$ (Wüst 1920) [25]. Approximating $\ln x \approx x - 1$, a comparison with (32) shows that

$$\frac{D_f}{D_q} \approx q^{\text{sat}}(T). \tag{36}$$

Accordingly, along with global warming, the evaporation rates (35) of climate models increase exponentially in comparison with (32), based on the Clausius–Clapeyron formula for the saturated vapor pressure. This bias may contribute to the fact that latest climate models systematically underestimate the warming of the oceans (Weller et al., 2022) [17], perhaps by accelerating the hydrological cycle, even though observation suggests and most models assume that relative humidity at the sea surface is largely unaffected by global warming (Rapp 2014; Feistel and Hellmuth 2021, 2023) [7,16,18].

## 4. Evaporation Enthalpy

Along with the mass flux of water evaporating from the oceans at a rate of about 1000 mm per year, the heat of solar irradiation is transferred as latent heat to the troposphere. The thermodynamic equation for the latent heat, $L^{\text{evap}}$, of the reversible evaporation of seawater into humid air was derived to read (Feistel et al., 2010; Feistel and Hellmuth 2023) [18,39]

$$L^{\text{evap}} = h^{\text{AV}} - A\left(\frac{\partial h^{\text{AV}}}{\partial A}\right)_{T,p} - h^{\text{SW}} + S\left(\frac{\partial h^{\text{SW}}}{\partial S}\right)_{T,p}. \tag{37}$$

Let $m_W, m_S, m_V,$ and $m_A$, respectively, be the partial masses of liquid water, dissolved sea salt, water vapor, and dry air involved in the evaporation process. In Equation (37), $h^{\text{AV}}(A, T, p)$ is the specific enthalpy of humid air as a function of the mass fraction $A = m_A/(m_V + m_A)$ of dry air, temperature $T$, and pressure $p$. The specific (or absolute) humidity is $q = 1 - A$. Similarly, $h^{\text{SW}}(S, T, p)$ is the specific enthalpy of seawater as a function of the mass fraction $S = m_S/(m_W + m_S)$ of salt dissolved, also known as absolute (or specific) salinity of IAPSO Standard Seawater with Reference Composition (Millero et al., 2008) [82]. Evaporation is reversible (without production of entropy) if the chemical potentials of water in seawater, Equation (21), and of water vapor in humid air, Equation (22), take equal values so that the phase equilibrium condition has the form

$$0 = g^{\text{AV}} - A\left(\frac{\partial g^{\text{AV}}}{\partial A}\right)_{T,p} - g^{\text{SW}} + S\left(\frac{\partial g^{\text{SW}}}{\partial S}\right)_{T,p}. \tag{38}$$

The specific entropies are related to the Gibbs energies and the enthalpies by (Feistel et al., 2010) [39]

$$\eta^{\text{AV}} = -\left(\frac{\partial g^{\text{AV}}}{\partial T}\right)_{A,p} = \frac{1}{T}\left(h^{\text{AV}} - g^{\text{AV}}\right) \tag{39}$$

and

$$\eta^{\text{SW}} = -\left(\frac{\partial g^{\text{SW}}}{\partial T}\right)_{S,p} = \frac{1}{T}\left(h^{\text{SW}} - g^{\text{SW}}\right) \tag{40}$$

so that the entropy of reversible evaporation is

$$K^{\text{evap}} = \eta^{\text{AV}} - A\left(\frac{\partial \eta^{\text{AV}}}{\partial A}\right)_{T,p} - \eta^{\text{SW}} + S\left(\frac{\partial \eta^{\text{SW}}}{\partial S}\right)_{T,p} = \frac{L^{\text{evap}}}{T}. \tag{41}$$

However, the typical sea surface relative humidity is about 80 %rh (Rapp 2014; Feistel and Hellmuth 2021) [7,16], and the related evaporation is an irreversible process accompanied by entropy production (Feistel and Ebeling 2011 [54]; Feistel 2019 [57]: Appendix C therein). The question of how large the related errors are if the reversible Equations (37) and (41) are applied to irreversible evaporation, i.e., if Equation (38) is violated, arises.

At constant $T$ and $p$, the total enthalpy of a two-box system consisting of one sample with seawater and one with humid air, such as that displayed in Figure 2, is

$$H(m_W, m_S, m_V, m_A, T, p) = (m_W + m_S)h^{SW}(S, T, p) + (m_V + m_A)h^{AV}(A, T, p). \quad (42)$$

If a mass portion of water, $dm$, is transferred from the seawater box to the air box, the latent heat of isobaric–isothermal evaporation is given by the implied change of total enthalpy

$$L^{evap} = \frac{1}{dm}[H(m_W - dm, m_S, m_V + dm, m_A, T, p) - H(m_W, m_S, m_V, m_A, T, p)], \quad (43)$$

that is,

$$L^{evap} = -h^{SW}(S, T, p) - (m_W + m_S)\left(\frac{\partial h^{SW}}{\partial S}\right)_{T,p}\left(\frac{\partial S}{\partial m_W}\right)_{m_S} + h^{AV}(A, T, p) + (m_V + m_A)\left(\frac{\partial h^{AV}}{\partial A}\right)_{T,p}\left(\frac{\partial A}{\partial m_V}\right)_{m_A}, \quad (44)$$

or, rearranged,

$$L^{evap}(A, S, T, p) = -h^{SW}(S, T, p) + S\left(\frac{\partial h^{SW}}{\partial S}\right)_{T,p} + h^{AV}(A, T, p) - A\left(\frac{\partial h^{AV}}{\partial A}\right)_{T,p}. \quad (45)$$

This formula is independent of any interaction between the two boxes as well as of the kind of water transfer mechanism involved. It is concluded that the agreement of Equation (45) with Equation (37) shows that the formula (37) for the reversible evaporation enthalpy as considered by Feistel and Hellmuth (2023) [18] is valid also for irreversible evaporation at mutually independent values of the four given variables, $A, S, T,$ and $p$.

## 5. Evaporation Entropy

At constant $T$ and $p$, the total entropy $N$ of a two-box system consisting of a sample with seawater and one with humid air is

$$N(m_W, m_S, m_V, m_A, T, p) = (m_W + m_S)\eta^{SW}(S, T, p) + (m_V + m_A)\eta^{AV}(A, T, p). \quad (46)$$

If a mass $dm$ of water evaporates, the entropy increases at a rate of

$$K^{evap} = \frac{1}{dm}[N(m_W - dm, m_S, m_V + dm, m_A, T, p) - N(m_W, m_S, m_V, m_A, T, p)], \quad (47)$$

which, similar to Equation (45), leads to

$$K^{evap} = -\eta^{SW}(S, T, p) + S\left(\frac{\partial \eta^{SW}}{\partial S}\right)_{T,p} + \eta^{AV}(A, T, p) - A\left(\frac{\partial \eta^{AV}}{\partial A}\right)_{T,p}. \quad (48)$$

Making use of Equations (39) and (40), this formula results in

$$TK^{evap} = L^{evap} + \Delta\mu_W. \quad (49)$$

Here, the quantity

$$\Delta\mu_W = \mu_W^{SW} - \mu_W^{AV} = g^{SW} - S\left(\frac{\partial g^{SW}}{\partial S}\right)_{T,p} - g^{AV} + A\left(\frac{\partial g^{AV}}{\partial A}\right)_{T,p} \quad (50)$$

is the difference between the chemical potential of water in seawater and that of water in humid air, that is, their distance from mutual chemical equilibrium. This distance can be expressed in terms of the relative fugacity, $\psi_f$, of humid air, Equations (29) and (32),

$$\Delta\mu_W \approx R_W T \left\{ \ln\left[1 + \frac{SM_W}{(1-S)M_S}\right] - \ln\psi_f(x_V, T, p) \right\}. \tag{51}$$

Here, the relative fugacity of humid air is (Feistel et al., 2015; Feistel and Lovell-Smith 2017) [52,83]

$$\psi_f(x_V, T) \equiv \frac{f_V^{AV}(x_V, T)}{f_V^{AV}(x_V^{sat}, T)}, \tag{52}$$

where $x_V^{sat}$ is the water vapor mole fraction of saturated humid air. For ideal gases, the relative fugacity equals the conventional relative humidity (Lovell-Smith et al., 2016) [85], $\psi_f \approx x_V / x_V^{sat}$.

According to Equation (49), the entropy of irreversible evaporation is, in contrast with Equation (41),

$$K^{evap}(A, S, T, p) = \frac{L^{evap}(A, S, T, p)}{T} + R_W \left\{ \ln\left[1 + \frac{SM_W}{(1-S)M_S}\right] - \ln\psi_f(x_V, T, p) \right\}. \tag{53}$$

Note that the additional irreversible contribution has a negative sign if the air above the sea surface (with salinity $S > 0$) is almost or fully saturated,

$$\left[1 + \frac{SM_W}{(1-S)M_S}\right]^{-1} < \psi_f \leq 1. \tag{54}$$

This means that in such a case, the mass flux is directed from the atmosphere to the ocean; that is, evaporation is replaced then by the condensation of water vapor.

The sea air entropy flux $J_N$ can be split into a reversible part, $J_N^{rev} \equiv J_W L^{evap}/T$, and the remaining irreversible part, $J_N^{irrev}$,

$$J_N = K^{evap} J_W = J_N^{rev} + J_N^{irrev}. \tag{55}$$

Irreversible evaporation means that along with the mass flux density $J_W$ of water, the ocean exports entropy at the rate $J_N^{rev}$, while the atmosphere imports entropy at the larger rate $J_N^{rev} + J_N^{irrev}$. This is reversed during irreversible condensation, when the ocean receives more entropy than the atmosphere is losing by this process.

Note that in the literature, the specific latent heat of vaporization is sometimes given by the difference of the specific entropies $\eta^W$ and $\eta^V$ of pure liquid water and water vapor, respectively, such as (Gill 1982 [31]: Equation 3.4.1 therein; Quasem et al., 2023 [86]: Equations 6.1 and 6.5 therein)

$$L \approx T\left(\eta^V - \eta^W\right). \tag{56}$$

This expression neglects the molecular interaction between water and either sea salt or dry air, and it applies only to saturated vapor that is formed reversibly, without producing additional entropy as it typically occurs at the ocean surface where the relative humidity deviated from its equilibrium value.

For an assessment of the relevance of the irreversible contribution to Equation (53), the ocean–atmosphere entropy flux density of evaporation may be considered in comparison with a mean global entropy export of about 1 W m$^{-2}$ K$^{-1}$ from the top of the atmosphere (Ebeling and Feistel 1982) [87], which corresponds to an atmospheric cooling rate of about 2 K day$^{-1}$ (Feistel and Ebeling 2011 [54]: Section 3.4 therein). Assuming an oceanic evaporation mass flux of Equation (32) with a Dalton constant of $D_f \approx 0.2$ g m$^{-2}$s$^{-1}$ (Feistel and Hellmuth 2023 [18]: Equations 22, 24 therein), a temperature of $T = 300$ K, a relative humidity of $\psi_f = 80$ %rh, and a salinity of $S = 35$ g kg$^{-1}$, the related fluxes

are estimated to be approximately $J_W \approx 0.04$ g m$^{-2}$ s$^{-1}$, $J_N^{\text{rev}} \approx 0.33$ W m$^{-2}$ K$^{-1}$, and $J_N^{\text{irrev}} \approx 0.004$ W m$^{-2}$ K$^{-1}$ (Feistel and Ebeling 2011: Equation 3.98 therein). Thus, in this rule-of-thumb estimate, the entropy produced by irreversible evaporation contributes only about 0.4% to the global entropy export of the earth, while the reversible flux amounts to about one-third. This reversible flux originates mainly from the entropy production occurring during the absorption of high-temperature solar irradiation by the comparatively cold terrestrial ocean (Feistel and Ebeling 2011; Feistel 2011) [54,79,88]. An alternative, more careful estimate of the oceanic entropy flux by Yan et al. (2004) [89] amounts to almost 0.6 W m$^{-2}$ K$^{-1}$.

## 6. Summary

Derived from Gibbs' (1873) [90] fundamental concept of thermodynamic potentials, the main result of the paper is given by the thermodynamically rigorous equations for the evaporation enthalpy and the evaporation entropy of seawater. Those are derived mathematically rather than empirically from related measurements (which these authors are not aware of to exist at all). These equations are intended for use as state-of-the-art quantitative reference equations for the various empirical formulas employed in modern numerical models. For example, the TEOS-10 SIA library (www.teos-10.org) and several IAPWS documents (IAPWS AN6-16 2016) [49] provide various double-precision numerical check values for seawater properties for this purpose. This summary repeats the key equations derived before and explains these novel fundamental equations describing nonequilibrium seawater evaporation. As indicated below, the summary's first equation repeats and explains Equation (32) derived before in the main part, the second equation does so for Equation (45), the third one for Equation (53), and the last one for Equation (56).

Owing to the difference in chemical potentials of water between seawater and humid air, evaporation of water from the ocean surface is an irreversible process. Regardless of that, in the literature, the latent heat of vaporization is mostly derived from equilibrium thermodynamics, often even approximately from the equilibrium between pure liquid water and pure saturated vapor. In this paper, the formalism of linear irreversible thermodynamics is applied to a conceptual evaporation model. Assuming local equilibrium conditions in the ocean and in the troposphere, thermodynamic equations provided by the geophysical standard TEOS-10 may be exploited, in particular by using mutually consistent equations for enthalpy, entropy, and chemical potentials of seawater and humid air without ideal gas approximations being required. The resulting form of the Dalton Equation (32) for the evaporation mass flux,

$$J_W = D_f \left\{ \ln\left[1 + \frac{SM_W}{(1-S)M_S}\right] - \ln\psi_f(x_V, T, p) \right\},$$

is expressed in terms of relative fugacity of humid air, $\psi_f$, which is a real-gas-corrected version of conventional definitions of relative humidity. The new Dalton equation predicts constant evaporation rates at constant sea surface relative humidity, in contrast with Dalton equations implemented in most climate models, which suggest an accelerated hydrological cycle under those conditions, implying numerical ocean cooling.

From the entropy balance at the air–sea interface, the equation for the latent heat of evaporation, Equation (45),

$$L^{\text{evap}} = -h^{\text{SW}}(S, T, p) + S\left(\frac{\partial h^{\text{SW}}}{\partial S}\right)_{T,p} + h^{\text{AV}}(A, T, p) - A\left(\frac{\partial h^{\text{AV}}}{\partial A}\right)_{T,p},$$

is demonstrated to be equally valid for reversible and irreversible evaporation. By contrast, the evaporation entropy, Equation (53),

$$K^{\text{evap}} = \frac{L^{\text{evap}}}{T} + R_W \left\{ \ln\left[1 + \frac{SM_W}{(1-S)M_S}\right] - \ln\psi_f(x_V, T, p) \right\},$$

includes an additional irreversible term proportional to the evaporation rate that vanishes at equilibrium. This new nonequilibrium term is missing in previous incomplete textbook definitions, such as Equation (56),

$$L \approx T\left(\eta^{\mathrm{V}} - \eta^{\mathrm{W}}\right),$$

for the latent heat of vaporization in terms of entropies of water and vapor; that is, the difference $\left(\eta^{\mathrm{V}} - \eta^{\mathrm{W}}\right)$ does not represent the evaporation entropy.

**Supplementary Materials:** The following supporting information can be downloaded at: https://www.mdpi.com/article/10.3390/jmse12010166/s1. Refs. [91–189] are cited in Supplementary Materials.

**Funding:** This research received no external funding.

**Acknowledgments:** The authors are grateful to Thomas Foken for providing the data shown in Figure 1 and various references and to Trevor McDougall and four anonymous reviewers for their additional suggestions. This article contributes to the tasks of the IAPSO/SCOR/IAPWS Joint Committee on the Properties of Seawater (JCS). The contribution of O. Hellmuth was provided within the framework of research theme 2 "Aerosols and clouds, long-term processes and trends" of the Leibniz Institute for Tropospheric Research (TROPOS), Leipzig, and is part of the TROPOS activities within the framework of the EU project "Aerosol, Clouds and Trace gases Research InfraStructure" (ACTRIS).

**Conflicts of Interest:** The authors declare no conflicts of interest.

## Nomenclature

| Symbol | Quantity | Basic SI Unit |
|---|---|---|
| $A$ | mass fraction of dry air in humid air, $A = m_{\mathrm{A}}/(m_{\mathrm{A}} + m_{\mathrm{V}})$ | $\mathrm{kg\,kg^{-1}}$ |
| $c$ | unit conversion constant | $\mathrm{J\,kg^{-1}K^{-1}}$ |
| $C_p$ | isobaric heat capacity | $\mathrm{J\,K^{-1}}$ |
| $D_f$ | fugacity-based Dalton coefficient | $\mathrm{kg\,m^{-2}\,s^{-1}}$ |
| $D_q$ | humidity-based Dalton coefficient | $\mathrm{kg\,m^{-2}\,s^{-1}}$ |
| $e$ | partial pressure of water vapor of humid air | Pa |
| $e^{\mathrm{sat}}$ | saturation vapor pressure of water | Pa |
| $f_{\mathrm{V}}^{\mathrm{AV}}$ | fugacity of water vapor in humid air | Pa |
| $f_{\mathrm{V}}^{\mathrm{id}}$ | ideal gas fugacity of water vapor | Pa |
| $f_{\mathrm{W}}^{\mathrm{SW}}$ | fugacity of water in seawater | Pa |
| $G$ | Gibbs energy (or free enthalpy), $G = H - TN$ | J |
| $g$ | specific Gibbs energy, Gibbs function | $\mathrm{J\,kg^{-1}}$ |
| $G^{\mathrm{AV}}$ | Gibbs energy of humid air | J |
| $g^{\mathrm{AV}}$ | specific Gibbs energy of humid air | $\mathrm{J\,kg^{-1}}$ |
| $G^{\mathrm{SW}}$ | Gibbs energy of seawater | J |
| $g^{\mathrm{SW}}$ | specific Gibbs energy of seawater | $\mathrm{J\,kg^{-1}}$ |
| $H$ | enthalpy, $H = U + pV$ | J |
| $h$ | specific enthalpy, $h = H/m$ | $\mathrm{J\,kg^{-1}}$ |
| $h^{\mathrm{AV}}$ | specific enthalpy of humid air | $\mathrm{J\,kg^{-1}}$ |
| $h^{\mathrm{SW}}$ | specific enthalpy of seawater | $\mathrm{J\,kg^{-1}}$ |
| $H^{*}$ | potential enthalpy | J |
| $J_k$ | any Onsager flux | |

| Symbol | Quantity | Basic SI Unit |
|---|---|---|
| $J_N$ | entropy flux | $\mathrm{W\,m^{-2}\,K^{-1}}$ |
| $J_N^{\text{irrev}}$ | irreversible entropy flux | $\mathrm{W\,m^{-2}\,K^{-1}}$ |
| $J_N^{\text{rev}}$ | reversible entropy flux | $\mathrm{W\,m^{-2}\,K^{-1}}$ |
| $J_W$ | evaporation mass flux of water | $\mathrm{kg\,m^{-2}\,s^{-1}}$ |
| $K^{\text{evap}}$ | specific entropy of evaporation | $\mathrm{J\,kg^{-1}K^{-1}}$ |
| $L$ | specific latent heat of vaporization | $\mathrm{J\,kg^{-1}}$ |
| $L^{\text{evap}}$ | specific latent heat of evaporation | $\mathrm{J\,kg^{-1}}$ |
| $\boldsymbol{m}$ | vector of masses | kg |
| $m$ | mass | kg |
| $M_A$ | molar mass of dry air, $M_A = 0.02896546\ \mathrm{kg\,mol^{-1}}$ | $\mathrm{kg\,mol^{-1}}$ |
| $m_A$ | mass of dry air in humid air | kg |
| $m_k$ | mass of substance $k$ | kg |
| $M_S$ | molar mass of sea salt, $M_S = 0.031403822\ \mathrm{kg\,mol^{-1}}$ | $\mathrm{kg\,mol^{-1}}$ |
| $m_S$ | mass of dissolved sea salt | kg |
| $m_V$ | mass of water vapor | kg |
| $M_W$ | molar mass of water, $M_W = 0.018015268\ \mathrm{kg\,mol^{-1}}$ | $\mathrm{kg\,mol^{-1}}$ |
| $m_W$ | mass of water solvent | kg |
| $N$ | entropy | $\mathrm{J\,K^{-1}}$ |
| $P$ | entropy production | $\mathrm{W\,K^{-1}}$ |
| $p$ | pressure | Pa |
| $p_0$ | reference pressure, surface pressure | Pa |
| $q$ | specific humidity, $q = 1 - A = m_V/(m_A + m_V)$ | $\mathrm{kg\,kg^{-1}}$ |
| $q^{\text{sat}}$ | saturation specific humidity | $\mathrm{kg\,kg^{-1}}$ |
| $R$ | molar gas constant, $R = 8.31446262\ \mathrm{J\,mol^{-1}K^{-1}}$ | $\mathrm{J\,mol^{-1}K^{-1}}$ |
| $R_W$ | specific gas constant of water, $R_W = R/M_W$ | $\mathrm{J\,kg^{-1}K^{-1}}$ |
| $S$ | absolute salinity, $S = m_S/(m_W + m_S)$ | $\mathrm{kg\,kg^{-1}}$ |
| $T$ | ITS-90 temperature | K |
| $t$ | time | s |
| $U$ | internal energy | J |
| $u$ | specific internal energy, $u = U/m$ | $\mathrm{J\,kg^{-1}}$ |
| $V$ | volume | $\mathrm{m^3}$ |
| $v$ | specific volume | $\mathrm{m^3\,kg^{-1}}$ |
| $\boldsymbol{w}$ | vector of mass fractions | |
| $w_k$ | mass fraction of substance $k$, $w_k = m_k/m$ | $\mathrm{kg\,kg^{-1}}$ |
| $\boldsymbol{X}$ | mass flux driving force | |
| $X_k$ | any Onsager force | |
| $\boldsymbol{X}_k$ | Onsager force vector of mass $m_k$ | $\mathrm{m\,s^{-2}K^{-1}}$ |
| $\boldsymbol{X}_u$ | Onsager force vector of internal energy | $\mathrm{m^{-1}K^{-1}}$ |
| $\boldsymbol{X}_v$ | Onsager force vector of volume | $\mathrm{Pa\,m^{-1}K^{-1}}$ |
| $x_V$ | mole fraction of water vapor in humid air | $\mathrm{mol\,mol^{-1}}$ |

| Symbol | Quantity | Basic SI Unit |
|--------|----------|---------------|
| $x_V^{\text{sat}}$ | saturation mole fraction of water vapor in humid air | $\text{mol mol}^{-1}$ |
| $X_W$ | Onsager force of water evaporation | $\text{J m}^{-1}\text{kg}^{-1}\text{K}^{-1}$ |
| $x_W$ | mole fraction of liquid water in seawater | $\text{mol mol}^{-1}$ |
| $\Delta z$ | laminar layer thickness | m |
| $\Theta$ | conservative temperature | K |
| $\vartheta$ | potential temperature | K |
| $\delta$ | Sverdrup's diffusion coefficient (characteristic time) | s |
| $\eta$ | specific entropy, $\eta = N/m$ | $\text{J kg}^{-1}\text{K}^{-1}$ |
| $\eta^{\text{AV}}$ | specific entropy of humid air | $\text{J kg}^{-1}\text{K}^{-1}$ |
| $\eta^{\text{SW}}$ | specific entropy of seawater | $\text{J kg}^{-1}\text{K}^{-1}$ |
| $\eta^{\text{V}}$ | specific entropy of water vapor | $\text{J kg}^{-1}\text{K}^{-1}$ |
| $\eta^{\text{W}}$ | specific entropy of liquid water | $\text{J kg}^{-1}\text{K}^{-1}$ |
| $\mu$ | chemical potential | $\text{J kg}^{-1}$ |
| $\boldsymbol{\mu}$ | vector of chemical potentials | $\text{J kg}^{-1}$ |
| $\mu_V^{\text{id}}$ | ideal gas chemical potential of water vapor | $\text{J kg}^{-1}$ |
| $\mu_W^{\text{id}}$ | ideal-gas chemical potential of liquid water | $\text{J kg}^{-1}$ |
| $\mu_k$ | chemical potential of substance $k$ | $\text{J kg}^{-1}$ |
| $\mu_V^{\text{AV}}$ | chemical potential of water vapor in humid air | $\text{J kg}^{-1}$ |
| $\mu_W^{\text{SW}}$ | chemical potential of water in seawater | $\text{J kg}^{-1}$ |
| $\Delta\mu_W$ | chemical potential difference of water | $\text{J kg}^{-1}$ |
| $\psi_f$ | relative fugacity | 1 |
| $\Omega_{ik}$ | Onsager coefficient | |
| $\Omega_{\text{WW}}$ | Onsager coefficient of water evaporation | $\text{kg K s m}^{-3}$ |

## Notes

[1] Halley (1687) [1]: p. 368.
[2] Dalton (1789) [2]: p. 537.
[3] Albrecht (1940) [8]: p. 36, 77.
[4] Randall (2012) [9]: p. 176.
[5] Albrecht (1940) [8]: p. 77.
[6] Rapp (2014) [16]: p. 420.
[7] Weller et al. (2022) [17]: p. E1968.
[8] Planck (1948) [74]: p. 28, 29.

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
