# Peer review of "Irreversible Thermodynamics of Seawater Evaporationâ€"

_jmse, doi:10.3390/jmse12010166_

Round 1
Reviewer 1 Report
Comments and Suggestions for Authors
Peer-review
Title: “Irreversible Thermodynamics of Seawater Evaporation”
by R. Feistel and O. Hellmuth
The peer-reviewed manuscript makes an ambiguous and rather strange impression. The authors, on the one hand, consider an interesting issue regarding the evaporation of water from the ocean surface into the atmosphere, which is important for climate modeling. On the other hand, the consideration of this issue resembles the material of a lecture for university students and not a scientific article with novelty. The article is heavily overloaded with auxiliary material and is very difficult to be understandable by readers due to the abundance of auxiliary and well-known mathematical expressions and formulas. If we neglect the known material from the manuscript, the question arises what contribution the authors made to the theory of evaporation? Moreover, the authors themselves did not clearly formulate the purpose of their work in the introduction, while the Summary does not withstand any criticism at all. It is not clear why a photograph of Ludwig Boltzmann’s grave is given in a scientific article?
In addition, the authors made some statements that seem plausible, but are not entirely substantiated. This concerns, for example, the assertion that the parameterization schemes used in climate models to describe the atmosphere-ocean interactions do not quite adequately describe evaporation processes and this is the reason that climate models are not able to estimate the rate of global warming with the necessary reliability. First of all, it is not clear what “with reasonable certainty” means? Secondly, there are much more important reasons behind the inter-model differences in climate change estimates. These reasons include, for example, scenarios of human-induced effects on the climate system.
Unfortunately, I cannot recommend the peer-reviewed manuscript for publication.
Reviewer 2 Report
Comments and Suggestions for Authors
This paper tries to discuss about the irreversible thermodynamics of seawater evaporation. The authors apply the formalism of linear irreversible thermodynamics to a conceptual model of seawater evaporation, assuming local equilibrium conditions in the ocean and in the troposphere. They use the thermodynamic equations provided by TEOS-10, the current international geophysical standard for thermodynamic properties of seawater, ice and humid air, to present the chemical potentials, enthalpies and entropies of water in both phases. They try to estimate the entropy production rate of ocean evaporation and compare it to the global entropy export of the Earth.
Figure 1 that the authors present here, having an introductory manner, I think should only be quoted in the text and detailed in the sense of motivation. I'm sure the authors realize that even the scale of the OX and OY axes is not represented in a scientific manner. However, here, I do not understand why authors do not use recent articles and books to create a state of the art as it lends itself to in a scientific article.
The authors did not follow the format of the journal, based on a template that can be easily downloaded from this web address: https://www.mdpi.com/journal/jmse/instructions. Of course, this has nothing to do with the quality of the paper, but I am sure that the authors also evaluate scientific articles and understand, for example, the importance of line numbering.
I don't see the sense in which the structure of the work is presented. This introductory chapter should motivate the study and present its originality. The authors present a modest state of the art. It should be correlated with more recent studies.
This article is rather a very good book chapter. It cannot be considered a scientific article. The authors present a complex and beautifully correlated theoretical study, but no results are presented by presenting correlations. The results are presented towards the end very briefly. I recommend authors to present them on a table. The conclusions, in fact the subchapter "Summary" is actually another theoretical subchapter that is not clearly presented what the conclusions actually are.
Reviewer 3 Report
Comments and Suggestions for Authors
I appreciate the effort that went into exploring the Irreversible Thermodynamics of Seawater Evaporation in this study, highlighting its importance to air-sea interaction.
I was curious about the derivation presented. It comes across as quite theoretical. Would it be possible to share any validation for these derivations?
Furthermore, based on the derivation, could you possibly provide an order of magnitude estimation regarding its potential impact on the heat budget? I'm interested in understanding how this might have significant accumulative effects when considered on a larger scale or in conjunction with other simulations.
On a side note, I noticed the paper incorporates several quotes from older sources. It's an interesting approach; however, it's somewhat distinct from what one might commonly find in contemporary scientific writing.
Reviewer 4 Report
Comments and Suggestions for Authors
Comments:
- How does the concept of entropy production relate to the evaporation of water from the ocean?
- Could you explain the significance of considering 80% relative humidity in studying irreversible ocean evaporation?
- What fundamental equations are derived for the latent heat of irreversible evaporation, and how do they contribute to our understanding of this process?
- How does linear irreversible thermodynamics play a role in the conceptual ocean evaporation model?
- What is the TEOS-10 (Thermodynamic Equation of Seawater) standard, and how is it applied in irreversible evaporation?
- Can you elaborate on the concept of local thermodynamic equilibrium and its role in this study?
- What are the implications of the new equations derived for the mass flux of evaporation?
- How do non-equilibrium enthalpies and entropies relate to the process of ocean evaporation?
- What does the estimated entropy production rate of 0.004 W m^(-2) K^(-1) signify in the context of ocean evaporation?
- How does this entropy production rate compare to the average terrestrial global entropy production?
- Are there practical applications or implications of this research in climate science or oceanography?
- Could the findings of this study have any relevance to understanding climate change or global warming?
- How do local equilibrium conditions affect the behavior of ocean evaporation, and why is this important to consider?
- Can you explain the methodology used to estimate the entropy production rate?
- Are there potential consequences or environmental impacts associated with entropy production during ocean evaporation?
- How might this research contribute to understanding heat exchange processes in the ocean-atmosphere system?
- What are the limitations or assumptions made in the study that researchers should be aware of?
- Are there any practical implications for industries or policies related to oceanic environments?
- How does this research fit into the broader field of thermodynamics and its applications?
- What future research directions or questions does this study raise in the context of ocean evaporation and entropy production?
- Could you provide some insights into the historical context or previous studies that led to this research?
- How does this research add to the body of knowledge regarding the Earth's energy balance?
- What motivated the choice of the specific entropy production rate units (W m^(-2) K^(-1)) for comparison?
- Are there any potential applications of these findings in renewable energy or desalination technologies?
- What role does entropy play in understanding the behavior of the ocean as a thermodynamic system?
- Can you summarize this study's key takeaways or implications for a non-expert audience?
Round 2
Reviewer 1 Report
Comments and Suggestions for Authors
Dear Authors,
Thanks a lot for clarification of some my questions.
Returning to my question about the purpose of your article, I would like to say the following. I understand what you are talking about in the article. However, anything can be considered, the main question is why it needs to be done. I'm not sure that using your parameterization the accuracy of climate projections will increase. The fact is that there are things that are much more important and which are characterized by a high degree of uncertainty.
Your work is very interesting and important theoretically. I support its publication. However, I ask you to make some small additions or changes.
Please, restate the purpose of the article in the introduction. Try to focus less on the climate issue. I believe that the use of the results of your article will be very useful in studying the processes of interaction between the atmosphere and the ocean on small spatial scales.
Reviewer 2 Report
Comments and Suggestions for Authors
I tried to access the http://www.teos-10.org/ site and it didn't work (not even on the first review). I now insisted on disabling the browser's security settings and managed to get through. I ask the authors to understand that an inaccessible cited website can raise questions nowadays, especially as it is not secure (https).
Regarding Figure 1, I was not careful enough in the first review to notice that the scale on the OY axis is in the order of millimetres. I thank the authors for their clarification and kindly ask them to accept my apology.
The authors have taken the observations into account and have completed the paper, which has been improved and presented more clearly on some aspects mentioned by the authors.
However, I ask the authors to re-state in chapter 6 what they wrote in their response letter. I therefore ask them to supplement the conclusions of this study.
Reviewer 4 Report
Comments and Suggestions for Authors
The authors have comprehensively addressed all the comments raised by the reviewer. The current version of the manuscript is now acceptable for publication.
